# PAM-flexible genome editing with an engineered chimeric Cas9

Lin Zhao [1], Sabrina R. T. Koseki[1], Rachel A. Silverstein[2,3,4], Nadia Amrani[5], Christina Peng [6], Christian Kramme[7], Natasha Savic[6], Martin Pacesa [8], Tomás C. Rodríguez [5], Teodora Stan [1], Emma Tysinger [1], Lauren Hong [1], Vivian Yudistyra [1], Manvitha R. Ponnapati[9], Joseph M. Jacobson[9], George M. Church[7], Noah Jakimo[9], Ray Truant [6], Martin Jinek [8], Benjamin P. Kleinstiver [2,3,10], Erik J. Sontheimer [5] & Pranam Chatterjee [1,11] ✉

CRISPR enzymes require a defined protospacer adjacent motif (PAM) flanking a guide RNA-programmed target site, limiting their sequence accessibility for robust genome editing applications. In this study, we recombine the PAM-interacting domain of SpRY, a broad-targeting Cas9 possessing an NRN > NYN (R = A or G, Y = C or T) PAM preference, with the N-terminus of Sc + +, a Cas9 with simultaneously broad, efficient, and accurate NNG editing capabilities, to generate a chimeric enzyme with highly flexible PAM preference: SpRYc. We demonstrate that SpRYc leverages properties of both enzymes to specifically edit diverse PAMs and disease-related loci for potential therapeutic applications. In total, the approaches to generate SpRYc, coupled with its robust flexibility, highlight the power of integrative protein design for Cas9 engineering and motivate downstream editing applications that require precise genomic positioning.

To conduct programmable genome editing, CRISPR-associated (Cas) endonucleases require a protospacer adjacent motif (PAM) to immediately follow the target DNA sequence specified by the guide RNA (gRNA)[1–3]. PAM binding triggers DNA strand separation, enabling base pairing between the gRNA and the target DNA strand for subsequent nucleolytic cleavage and editing events[4,5]. The widely-utilized Cas9 from *Streptococcus pyogenes* bacteria (SpCas9), for example, requires a 5'-NGG-3' PAM[2,6,7], imposing severe accessibility constraints for therapeutically relevant editing applications requiring precise genomic positioning, such as base editing and homology-directed repair[8–12].

To expand the targetable sequence space of CRISPR, we previously engineered Sc + +, a variant of ScCas9 which employs a positive-charged loop that relaxes the base requirement at the second PAM position, thus enabling a 5'-NNG-3' preference, rather than the canonical 5'-NGG-3'[13,14]. Concurrent with the development of Sc + +, Walton, et al. engineered a near-PAMless Cas9, termed SpRY, which contains mutations in the PAM-interacting domain (PID) of SpCas9 that enable strong 5'-NRN-3' specificity, alongside weaker 5'-NYN-3' targeting[15]. Both Sc + + and SpRY thus represent exciting advances in CRISPR-based genome editing due to their robust editing characteristics and unprecedented genomic accessibility, respectively[16].

In this study, we combine Sc + + and SpRY to engineer a chimeric Cas9 enzyme that can induce edits with orthogonal PAM targeting. To do this, we employ experimental enzyme engineering and computational modeling to graft the PID of SpRY to the N-terminus of Sc + +, generating a chimeric SpRY-Sc + + Cas9 (herein referred to as SpRYc).

[1]Department of Biomedical Engineering, Duke University, Durham, NC, USA. [2]Center for Genomic Medicine, Massachusetts General Hospital, Boston, MA, USA. [3]Department of Pathology, Massachusetts General Hospital, Boston, MA, USA. [4]Biological and Biomedical Sciences Program, Harvard University, Boston, MA, USA. [5]RNA Therapeutics Institute, University of Massachusetts Medical School, Cambridge, USA. [6]Department of Biochemistry and Biomedical Sciences, McMaster University, Hamilton, Canada. [7]Wyss Institute for Biologically Inspired Engineering, Harvard University, Cambridge, MA, USA. [8]Department of Biochemistry, University of Zurich, Zürich, Switzerland. [9]Media Lab, Massachusetts Institute of Technology, Cambridge, MA, USA. [10]Department of Pathology, Harvard Medical School, Boston, MA, USA. [11]Department of Computer Science, Duke University, Durham, NC, USA. ✉e-mail: pranam.chatterjee@duke.edu

We demonstrate that SpRYc integrates the loop structure of Sc + + and the PID mutations of SpRY to specifically edit various 5′-NNN-3′ PAM targets in human cells, enabling specific editing applications. Finally, we conduct homology modeling to gain insights into the protein-DNA interactions that may enable SpRYc's PAM flexibility. In total, SpRYc's demonstrated PAM flexibility offers numerous opportunities for broad-targeting genome editing applications and therapeutic translation.

## Results

### Engineering of SpRYc

SpRY harbors ten substitutions in the PID of SpCas9 (L1111R, D1135L, S1136W, G1218K, E1219Q, A1322R, R1333P, R1335Q, and T1337R) which help reduce its specificity from the canonical 5′-NGG-3′ to the more flexible 5′-NRN-3′ PAM[15]. Alternatively, ScCas9 and Sc + + both employ positive-charged, flexible loop-like structures in their N-terminus (residues 367 to 376) that do not exist in SpCas9 or SpRY, and relax the need for the second PAM base, enabling more minimal 5′-NNG-3′ PAM preference rather than 5′-NGG-3′[13,14].

Previously, we grafted the GC-independent PID of *Streptococcus macacae* Cas9 to the N-terminus of its ortholog, SpCas9, to generate iSpyMac, an efficient 5′-NAA-3′ editor[17]. Motivated by our previous domain grafting results, we engineered a single variant possessing the critical properties of SpRY and Sc + + by rationally exchanging the PID of Sc + + with that of SpRY to generate a chimeric hybrid Cas9: SpRYc. SpRYc consists of the N-terminus (residues 1–1119) of Sc + +, including the flexible loop, followed by the region of SpRY (residues 1111–1368) spanning its PID mutations (Fig. 1A).

### PAM characterization of SpRYc

To experimentally interrogate the PAM specificity of SpRYc in comparison to SpCas9, Sc + +, and SpRY, we adapted a positive selection bacterial screen based on green fluorescent protein (GFP) expression conditioned on PAM binding, termed PAM-SCANR[18]. Following the transformation of the PAM-SCANR plasmid, harboring a PAM library, a single gRNA (sgRNA) plasmid targeting the fixed PAM-SCANR protospacer, and a corresponding nuclease-deficient dCas9 plasmid, we conducted FACS analysis to isolate GFP-positive cells in each population for subsequent library amplification and sequencing (Supplementary Fig. 1). Our aggregate Sanger Sequencing results suggest that while SpRY preferentially binds to PAM sequences with an A or G at position 2, as expected, SpRYc more potently binds sequences with adenine bases at position 2 without bias against any specific base (Fig. 1B).

We performed additional experiments to assess PAM preference using HT-PAMDA, which calculates the cleavage rates of Cas9 enzymes (as opposed to an endpoint assay, like PAM-SCANR) on a library of substrates harboring different PAMs[19]. While we observe far broader editing capabilities of SpRYc than the 5′-NGG-3′ PAM of SpCas9 and the strong 5′-NNG-3′ PAM of Sc + +, SpRYc exhibited slower cleavage rates than SpRY, though able to access a comparably broad set of PAMs, thus suggesting that SpRYc may elicit its optimal activity in either its "dead" or nickase variants, rather than as a nuclease (Fig. 1C). Overall, these results motivated us to evaluate SpRYc's genome editing capability on endogenous loci in both nuclease and non-cleavage editing formats.

### Human genome editing capabilities of SpRYc

To evaluate SpRYc's activity at diverse gene sequences, we compared the PAM specificities and DNA cleavage capabilities of SpRYc to SpCas9 and SpRY by transfecting HEK293T cells with plasmids expressing each Cas9 alongside one of sixteen sgRNAs which were directed to various genomic loci representing every two-base PAM combination (5′-N**NN**−3′) (Supplementary Table 1). Five days after

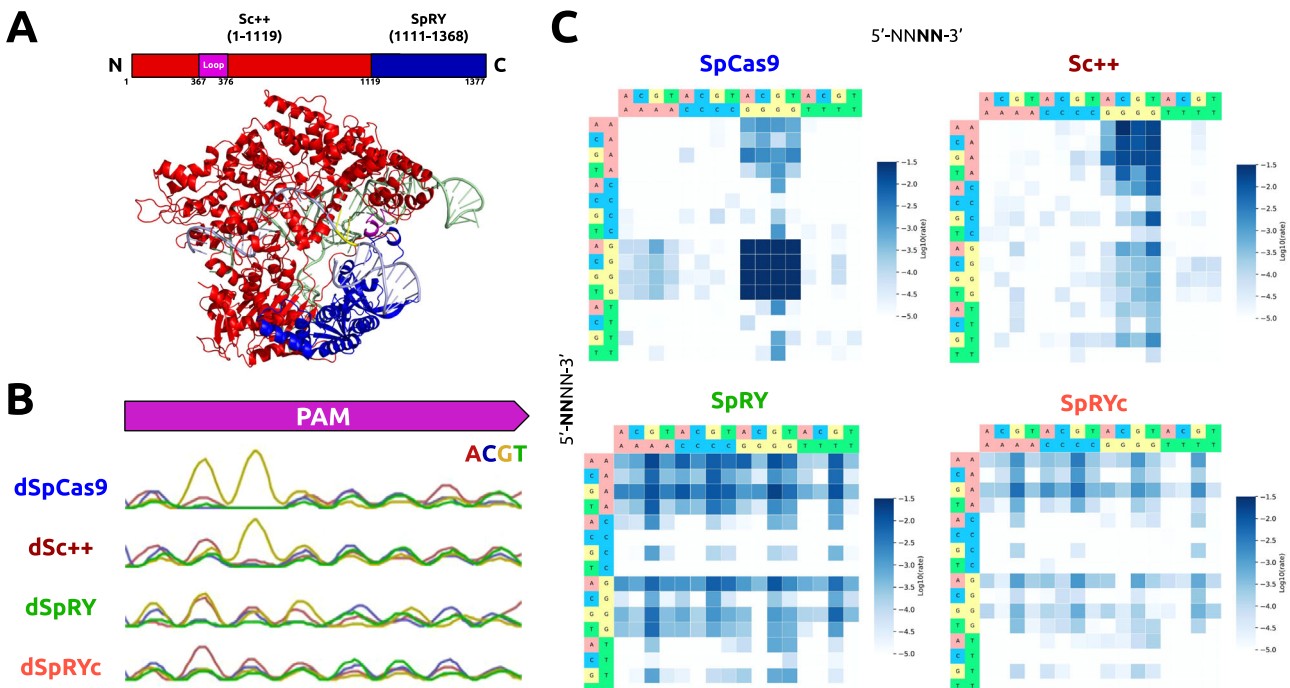

**Fig. 1 | Engineering, modeling, and PAM Characterization of SpRYc.**
**A** Homology model of SpRYc generated in SWISS-MODEL from PDB 4UN3 and visualized in PyMol. Original domain coordinates are indicated in parentheses above while SpRYc coordinates are indicated below. PAM is indicated in yellow, loop in purple, Sc + + N-terminus in red, and SpRY PID in blue. **B** PAM enrichment for indicated dCas9 enzymes utilizing PAM-SCANR. Each dCas9 plasmid was electroporated in duplicates, subjected to FACS analysis, and gated for GFP expression based on a negative "No Cas9" control and a positive dSpCas9 control. All samples were performed in independent transformation replicates, and the PAMs of the GFP-positive cells were sequenced via Sanger sequencing. **C** PAM profiles of SpCas9, Sc + +, SpRY, and SpRYc proteins as determined by HT-PAMDA. Rate constants corresponding to Cas cleavage activity are illustrated as log10 values and are the mean of cleavage reactions against two unique spacer sequences.

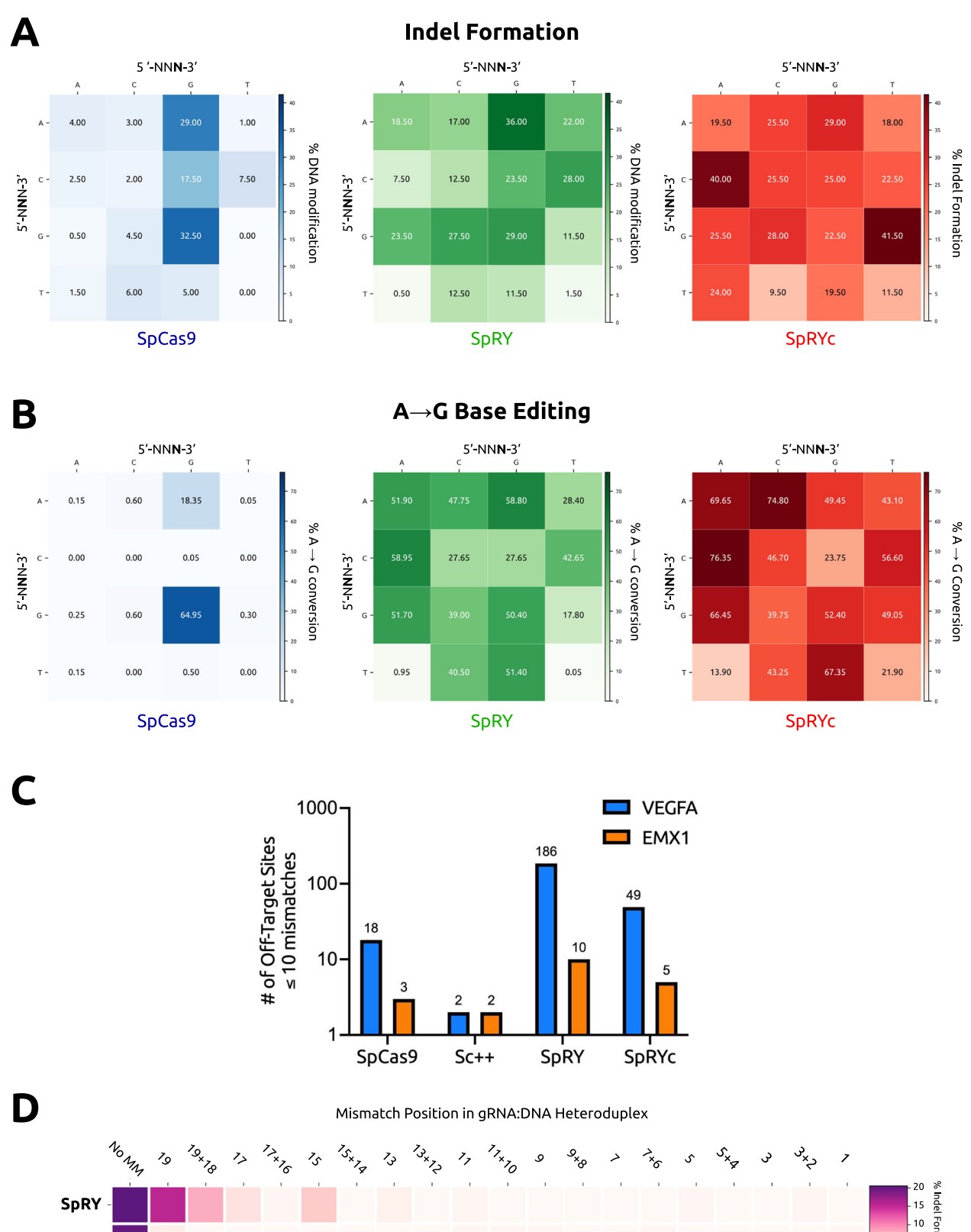

transfection, indel formation was quantified following PCR amplification of the target genomic regions and subsequent sequencing analysis. Our results demonstrate that SpRYc generates modifications at all tested genomic loci, performing comparably to SpRY, and more optimally on select 5′-NYN-3′ loci (Fig. 2A, Source Data). We similarly tested the performance of SpRYc in comparison to SpCas9 and SpRY

for base editing applications by fusing each variant to ABE8e, a rapid, high-activity adenine base editor[20,21]. We quantified the editing efficiency of the base with the highest conversion percentage in the editing window following PCR amplification of the target genomic regions. Our results reveal that SpRYc-ABE8e can base edit at all tested genomic sequences, as compared to SpCas9-ABE8e and on most of the

**Fig. 2 | Genome editing capabilities of SpRYc. A** Quantitative analysis of indel formation with indicated Cas9 variants. Indel frequencies were determined via batch analysis following PCR amplification of indicated genomic loci, in comparison to unedited controls for each gene target. All samples were performed in independent transfection replicates and the mean of the quantified indel formation values was calculated. All gRNA sequences can be found in Supplementary Table 1. **B** Quantitative analysis of A-to-G with indicated ABE8e variants. Base editing conversion rates were determined via BEEP following PCR amplification of indicated genomic loci, in comparison to unedited controls for each gene target. All samples were performed in independent transfection replicates and the mean of the

quantified base editing formation values was calculated. All gRNA sequences can be found in Supplementary Table 1. **C** Off-targets as identified by GUIDE-seq genome-wide for SpCas9, Sc + +, SpRY, and SpRYc each paired with two sgRNAs targeting either *EMX1* or *VEGFA*. Only sites that harbored a sequence with ≤10 mismatches relative to the gRNA were considered potential off-target sites. **D** Efficiency heat-map of mismatch tolerance assay on genomic targets. Quantified indel frequencies are exhibited for each labeled single or double mismatch (number of bases 5′ upstream of the PAM) in the sgRNA sequence for the indicated Cas9 variant and indicated PAM sequence. All samples were performed in independent transfection replicates and the mean of the quantified indel formation values was calculated.

assayed loci more optimally than SpRY-ABE8e. In particular, SpRYc-ABE8e greatly outperformed SpRY-ABE8e on 5′-N**T**N-3′ and 5′-NN**T**−3′ PAMs, such as 5′-NTT-3′, where SpRYc-ABE8e produced a 21.9% A to G conversion, compared to 0.05% with SpRY-ABE8e (Fig. 2B, Supplementary Fig. 2, Supplementary Tables 1 and 2). Taken together, our results suggest that SpRYc is able to target, cleave, and base edit at genomic sites with minimal dependence on a specific PAM.

### Reduced off-target propensity of SpRYc

Previously, we demonstrated that Sc + + is an intrinsically high-fidelity enzyme, with far reduced off-targeting as compared to the standard SpCas9[14]. We thus hypothesized that SpRYc may possess a lower off-target propensity than its SpCas9-derived counterpart, SpRY. To investigate this hypothesis, we employed the genome-wide, unbiased GUIDE-Seq method[22], by utilizing sgRNA sequences targeting two previously analyzed genomic loci (*VEGFA* and *EMX1*). Our results demonstrate that compared to SpRY, SpRYc has nearly four-fold lower off-target activity with the VEGFA-targeting guide RNA, and two-fold lower off-target activity when directed against the EMX1 site (Fig. 2C, Supplementary Figs. 3, 4). We corroborated this data via a mismatch tolerance assay[23], in which we employed sgRNAs harboring double or single mismatches to a fixed protospacer for an endogenous *DNMT1* locus. SpRYc exhibited decreased activity on mismatched sequences, as compared to SpRY, with slightly lowered on-target activity (Fig. 2D, Supplementary Fig. 5). Our observations thus support our HT-PAMDA results that SpRYc's attenuation in nuclease efficiency may result in fewer off-targets and improved mismatch tolerance[19].

### SpRYc base editors mediate therapeutically relevant edits

Having established SpRYc's relevant editing capabilities in human cells, we sought to investigate its utility as a potential therapeutic modality for the treatment of genetic diseases. Rett syndrome (RTT) is a progressive neurological disorder that predominantly affects young females. A majority of patients carry one of eight mutations in the *MECP2* gene (C316T, C397T, C473T, C502T, C763T, C808T, C880T, C916T), all of which are C-to-T substitution mutations and can thus be potentially ameliorated by CRISPR adenine base editors, such as ABE8e[12,20,24]. Notably, one of the eight mutations, C502T, can only be accessed at target sites consisting of a 5′-NCN-3′ or 5′-NTN-3′ PAM, preventing its correction by previous adenine base editors. To test whether SpRYc-ABE8e can effectively correct the C502T mutation, we generated a universal RTT HEK293T cell line via lentiviral-mediated, single-copy integration of a synthetic gene fragment encoding *MECP2* installed with the aforementioned RTT mutations. We transfected the SpRYc-ABE8e plasmid alongside an optimized sgRNA (5′-TGCTCTCACCGGGAGGGGCT-3′) for the C502T mutation site, harboring a previously-inaccessible 5′-CCC-3′ PAM (Fig. 3A, Supplementary Table 3). After subsequent DNA extraction, loci amplification, and next generation sequencing (NGS), we demonstrate that SpRYc-ABE8e can effectively correct *MECP2*, with an 34.73 ± 0.46% editing efficiency at the C502T mutation, comparable to that of SpRY-ABE8e's 34.43 ± 1.01% editing rate (Fig. 3B).

Huntington's Disease (HD) is a monogenic dominant neurological disorder affecting more than 1 in 10000 adults[25]. It is caused by an expanded CAG repeat on chromosome 4 of the *HTT* gene, which encodes an extended polyglutamine (polyQ) tract in the resulting *huntingtin* protein[25]. Recent studies have shown that there is an inverse relationship between the age of disease onset and the number of continuous CAG repeats, with the significant benefit of a natural interrupting CAA codon on age onset and severity of disease[26]. We, therefore, assessed SpRYc's ability to introduce silent CAA interruptions in the CAG repeat region of HTT. To do this, we transfected patient-derived TruHD fibroblast cells, possessing a clinically-relevant CAG repeat length of 43 repeats[27]. These lines are hTert immortalized, but not transformed and are very genomically stable. We used a cytosine base editor SpRYc-BE4Max alongside an sgRNA targeting the antisense strand of the *HTT* repeat region (Fig. 3C and Supplementary Table 3)[28]. Our combined NGS sequencing results show that SpRYc can install a CAA interruption at the fourth CAG repeat, with an A → G editing efficiency of 28.50 ± 5.31%, statistically comparable to that of SpRY-BE4Max's 22.90 ± 1.49%, thus shortening the uninterrupted repeat length by 4 and, for the expanded allele, reducing the CAG tract length to the sub-pathogenic range (Fig. 3C). Taken together, these results illustrate SpRYc's potential utility for clinically-relevant applications and motivate its potential development as a therapeutic platform.

### In silico modeling of SpRYc

To gain insights into the mechanisms of SpRYc's PAM targeting, and owing to the nearly 90% sequence similarity between ScCas9 and SpCas9, we conducted homology modeling of SpRYc in the DNA substrate bound-state using the SWISS-MODEL server (Figs. 1A and 3D)[29]. We hypothesized that the optimized loop of Sc + + may enforce targeting breadth by generating sequence-nonspecific interactions with the PAM to relax the need for an A or G at position 2. Homology models indicate that the engineered positively-charged loop inserted into the REC1 domain points towards the PAM region of the target DNA strands and thus potentially establishes new compensating interactions with the phosphate backbone of the target strand (Fig. 3Di). In addition, the combination of ScCas9 and SpRY mutations suggests several new nonspecific backbone interactions with the non-target strand, thereby supporting a relaxed PAM profile of SpRYc (Fig. 3Dii, iii). Of note is a potential van der Waals interaction of the aromatic side chain of W1145 with the ribose moieties of the proximal non-target strand residues (Fig. 3Div)[30]. These interactions, resulting from the engineered mutations, may thus energetically compensate for the lack of PAM-specific recognition and facilitate local unwinding of double-stranded DNA necessary for efficient R-loop formation in the absence of canonical PAM interactions. Finally, we demonstrate that SpRYc's interactions do not induce self-editing of gRNA plasmids, a potential complication of PAM-flexible CRISPR enzymes (Supplementary Fig. 6).

## Discussion

While PAMs play a critical role in self-nonself discrimination by pro-karyotic CRISPR-Cas9 immune systems, they limit the accessible

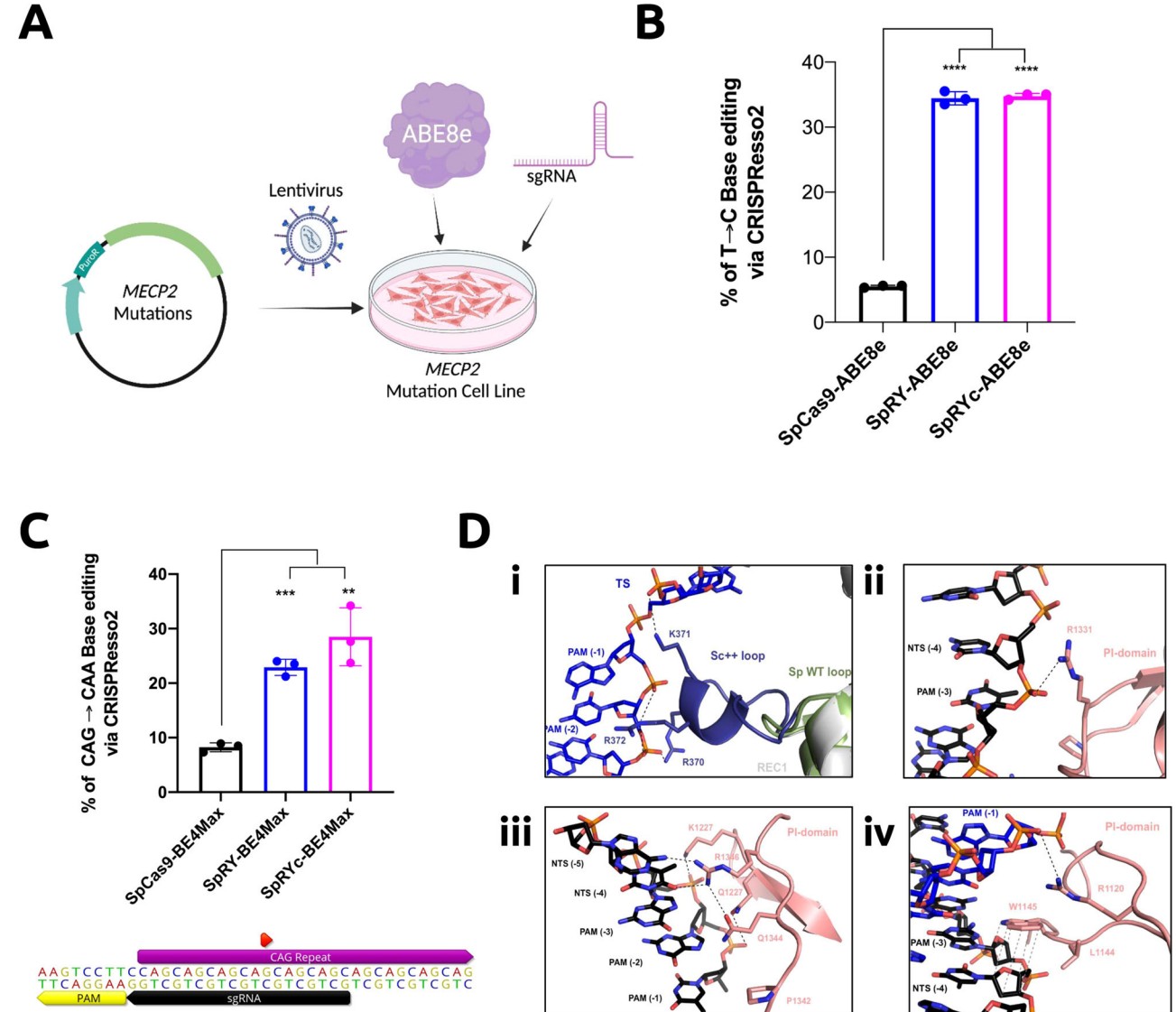

**Fig. 3 | Potential disease-associated loci editing applications and structural mechanisms of SpRYc. A** Schematic of SpRYc *MECP2* cell line generation and SpRYc-ABE8e editing. Figure was created with BioRender.com. **B** Base editing conversion rates were determined via CRISPResso2 NGS analysis following PCR amplification of *MECP2*-integrated loci, in comparison to SpCas9-ABE8e and SpRYc-ABE8e for the C502T installed mutation. Samples were performed in independent nucleofection triplicates ($n = 3$) ± SD, with the center of error bars depicting the mean. For individual samples, statistical significance was determined by a two-tailed Student's *t* test, as compared to the SpCas9-ABE8e control. The exact *p* values for SpRY-ABE8e and SpRYc-ABE8e compared to SpCas9-ABE8e are both <0.0001. Calculated *p* values are represented as follows: *, $p < 0.05$; **, $p < 0.01$; ***, $p < 0.001$; ****, $p < 0.0001$; ns, not significant. **C** SpRYc-BE4Max was nucleofected into TruHD cells alongside an sgRNA targeting the *HTT* repeat. Base editing conversion rate was determined via CRISPResso2 NGS analysis following PCR amplification of the *HTT* loci. Samples were performed in independent nucleofection triplicates ($n = 3$) ± SD, with the center of error bars depicting the mean. For

individual samples, statistical significance was determined by a two-tailed Student's *t* test, as compared to the SpCas9 control. The exact *p* values for SpRY-BE4Max and SpRYc-BE4Max compared to SpCas9-BE4Max are 0.001 and 0.0028, respectively. Calculated *p* values are represented as follows: *, $p < 0.05$; **, $p < 0.01$; ***, $p < 0.001$; ****, $p < 0.0001$; ns, not significant. The sgRNA and PAM sequences are annotated within their relative positions to the CAG repeat. The red annotation indicates the base to be mutated. The figure was made via Geneious Prime 2023.1.2. **D** Structural insights via homology modeling in SWISS-MODEL. (i) Interaction of the engineered Sc ++ loop (purple) with the backbone of the target strand (TS) PAM region. The REC1 loop from wild-type SpCas9 is indicated in green. (ii) Potential interaction of residue R1331 with the non-target strand (NTS) backbone. (iii) Multiple mutations within the PAM interaction loop allow for a more flexible PAM readout. (iv) The potential van der Waals interaction of W1145 with the ribose moieties of nontarget strand residues could further stabilize the PAM interaction.

sequence space for genome editing applications. Recent engineering and discovery efforts have yielded a host of Cas9 variants with altered or relaxed PAMs[13–15,17,31–37]. In this study, we engineer a chimeric Cas9 by harnessing the structural properties of SpRY and Sc ++ to generate SpRYc, a Cas9 with flexible PAM preference. While SpRYc did not demonstrate high cleavage rates in our HT-PAMDA assays, we do show that SpRYc has strong editing rates on diverse genomic loci as well as reduced off-target effects. SpRYc thus may be optimally fit for non-

nuclease editing applications, including precise base editing, prime editing, and CRISPR-mediated activation or inhibition. Further, due to the high sequence homology of ScCas9 and SpCas9, we anticipate that high-fidelity mutations[23,38,39] can easily be ported into SpRYc for improved specificity, as has been shown previously for both Sc ++ and SpRY[14,15]. Finally, we demonstrate that SpRYc can be integrated within base editing architectures to edit disease-related loci for potential therapeutic purposes.

While SpRYc serves as a step forward towards unrestricted, fully programmable genome editing, its development, more importantly, represents a culmination of a variety of state-of-the-art in silico and in vitro PAM engineering methods. ScCas9 was first identified via a high-throughput bioinformatics algorithm for ortholog discovery, dubbed SPAMALOT[13]. Its derivative, Sc++, was engineered by computationally identifying and extracting motifs from *Streptococcus* orthologs, and splicing them into ScCas9 for improved functionality[14]. Concurrently, SpRY was the result of a multi-year effort of SpCas9-based directed evolution and rational mutagenesis[15,31]. Finally, a combination of structure-based homology modeling and domain grafting methods, those that were instrumental in engineering other PAM variants such as iSpyMac[17] and cCas9[37], enabled the fusion of SpRY and Sc++ into our final SpRYc variant. Together, these studies emphasize the power of leveraging diverse engineering modalities to generate new and useful proteins and open the door for future integrative protein design.

## Methods

### Generation of plasmids
To generate SpRYc, the N-terminal ORF of Sc++ (Addgene Plasmid #155011), corresponding to residues (1–1119) was PCR amplified and assembled using Gibson Assembly into the pCMV-T7-SpRY-P2A-EGFP backbone (Addgene Plasmid #139989), preserving residues 1111–1368 of SpRY's ORF. pCMV-T7-SpCas9-P2A-EGFP (Addgene Plasmid #139987) was used for SpCas9, and Sc++ was similarly integrated within the backbone. Analogously, the ORFs of SpCas9, SpRY, and SpRYc were integrated within the ABE8e (Addgene Plasmid #138489), AncBE4Max (Addgene Plasmid #112094), or pCAG-CBE4max-SpRY-P2A-EGFP (Addgene Plasmid #139999) backbones, enforcing a D10A mutation. sgRNA plasmids were constructed by annealing oligonucleotides coding for crRNA sequences (Supplementary Fig. 1, Supplementary Table 1), as well as 4 bp overhangs, and subsequently performing a T4 DNA Ligase-mediated ligation reaction into a plasmid backbone immediately downstream of the human U6 promoter sequence. The *MECP2* editing locus containing all common Rett syndrome mutations was synthesized as a gBlock from IDT and inserted via Gibson cloning to a lentiviral vector harboring puromycin resistance. Assembled constructs were transformed into 50 μL NEB Turbo Competent *E. coli* cells, and plated onto LB agar supplemented with the appropriate antibiotic for subsequent sequence verification of colonies and plasmid purification.

### HT-PAMDA
We performed HT-PAMDA as described previously[19]. Briefly, HEK293T cells were transfected with plasmids encoding Cas9 nuclease variants, and in vitro cleavage assays were performed using the resulting cell lysates. sgRNAs were generated from Addgene plasmid #160136 with the T7 RiboMAX Express Large Scale RNA Production System (Promega). 180 ng of PAM library (Addgene #160132) was incubated with 30 nM of sgRNA and 6 μL of fluorescein-normalized lysate. PAM depletion was quantified following NGS of PCR-amplified undigested target DNA via the PAMDA software package: https://github.com/kleinstiverlab/HT-PAMDA. Cleavage rates for each Cas9 for each 5′-NNNN-3′ PAM can be accessed in the Source Data file.

### PAM-SCANR assay
Plasmids for the SpCas9 sgRNA and PAM-SCANR genetic circuit, as well as BW25113 ΔlacI cells, were generously provided by the Beisel Lab (North Carolina State University). Plasmid libraries containing the target sequence followed by either a fully-randomized 8-bp 5′-NNNNNNNN-3′ library or fixed PAM sequences were constructed by conducting site-directed mutagenesis, utilizing the KLD enzyme mix (NEB) after plasmid amplification, on the PAM-SCANR plasmid flanking the protospacer sequence (5′-CGAAAGGTTTTGCACTCGAC-3′).

Nuclease-deficient mutations (D10A and H850A) were introduced to the ScCas9 variants using Gibson Assembly as previously described. The provided BW25113 cells were made electrocompetent using standard glycerol wash and resuspension protocols. The PAM library and sgRNA plasmids, with resistance to kanamycin (Kan) and carbenicillin (Crb) respectively, were co-electroporated into the electrocompetent cells at 2.4 kV, outgrown, and recovered in Kan+Crb Luria Broth (LB) media overnight. The outgrowth was diluted 1:100, grown to ABS600 of 0.6 in Kan+Crb LB liquid media, and made electrocompetent. Indicated dCas9 plasmids, with resistance to chloramphenicol (Chl), were electroporated in duplicates into the electrocompetent cells harboring both the PAM library and sgRNA plasmids, outgrown, and collected in 5 mL Kan+Crb+Chl LB media. Overnight cultures were diluted to an ABS600 of 0.01 and cultured to an OD600 of 0.2. Cultures were analyzed and sorted on a FACSAria machine (Becton Dickinson). Events were gated based on forward scatter and side scatter and fluorescence was measured in the FITC channel (488 nm laser for excitation, 530/30 filter for detection), with at least 10,000 gated events for data analysis. Sorted GFP-positive cells were grown to sufficient density, plasmids from the pre-sorted and sorted populations were isolated, and the region flanking the nucleotide library was then PCR amplified and submitted for Sanger sequencing or Amplicon-EZ NGS analysis (Genewiz). FCS files were analyzed using FCSalyzer https://sourceforge.net/projects/fcsalyzer/, and gating strategy is described in Supplementary Fig. 1.

### Cell culture and DNA modification analysis
HEK293T cells (ATCC CRL-3216) were maintained in DMEM supplemented with 100 units/ml penicillin, 100 mg/ml streptomycin, and 10% fetal bovine serum (FBS). sgRNA plasmids (100 ng) and nuclease plasmids (100 ng) were transfected into cells as duplicates (2 ×10⁴ / well in a 96-well plate) with Lipofectamine 3000 (Invitrogen) in Opti-MEM (Gibco). Five days after transfection, genomic DNA was extracted using QuickExtract Solution (Lucigen), and genomic loci were amplified by PCR utilizing the Phusion Hot Start Flex DNA Polymerase (NEB). Amplicons were enzymatically purified and submitted for Sanger sequencing or Amplicon-EZ NGS sequencing (Genewiz). Sanger sequencing ab1 files were analyzed using the ICE web tool for batch analysis (ice.synthego.com)[40] in comparison to an unedited control to calculate indel frequencies via the ICE-D score. Select samples were further verified using the TIDE algorithm (tide.deskgen.com) to ascertain consistency of editing rates between replicates[41]. NGS FASTQ files were analyzed using a batch version of the software CRISPResso2 (https://github.com/pinellolab/CRISPResso2)[42]. Base editing files were analyzed via the Based Editing Evaluation Program (BEEP) (https://github.com/mitmedialab/BEEP) and Base Editing Analysis Tool (BEAT) (https://hanlab.cc/beat/) in comparison to an unedited control. All samples were performed in independent duplicates or triplicates, as indicated.

### GUIDE-Seq
We performed GUIDE-Seq as described previously[22]. Briefly, HEK293T cells (ATCC CRL-3216) were electroporated in a 24-well plate with 500 ng of Cas9, 500 ng of sgRNA, 10 ng of mCherry plasmids, and 7.5 pmol of annealed GUIDE-Seq oligonucleotide using the Neon nucleofection system (Thermo Fisher Scientific). After 72 h post-nucleofection, genomic DNA was extracted with a DNeasy Blood and Tissue kit (Qiagen 69504) according to the manufacturer's protocol. DNA libraries were prepared using custom oligonucleotides described in Tsai, et al.[22]. Library preparations were done with original adaptors with each library barcoded for pooled sequencing. The barcoded, purified libraries were sequenced on a MiniSeq platform in a paired-end (150/150) run.

Raw sequencer output (BCL) was demultiplexed and aligned to hg38 using GS-Preprocess (github.com/umasstr/GS-Preprocess)[43].

This software also constructed a reference of UMIs unique to each read and merged technical replicate BAM files. Off-target analysis of this input was performed using the GUIDEseq Bioconductor package[44]. Only sites that harbored a sequence with ≤10 mismatches relative to the gRNA were considered potential off-target sites. GUIDE-Seq read count data is indicated in Supplementary Figs. 3, 4.

## Lentiviral production

HEK293T cells (ATCC CRL-3216) were seeded in a 6-well plate and transfected at ~50% confluency. For each well, 0.5 μg pMD2.G (Addgene #12259), 1.5 μg psPAX2 (Addgene #12260) and 0.5 μg of the MECP2 vector were transfected with Lipofectamine 3000 (Invitrogen) according to the manufacturer's protocol. The medium was exchanged 8 h post transfection, and the viral supernatant was harvested at 48 and 72 h post-transfection. The viral supernatant was concentrated to 100x in 1x DPBS using Lenti-X Concentrator (Clontech, 631232) according to the manufacturer's instruction, and stored at −80 °C for further use.

## Rett syndrome cell line generation

$1 \times 10^5$ HEK293T cells (ATCC CRL-3216) were mixed with 20 μL of the concentrated virus in a 6-well plate. Media was changed 24 h post-transduction. Antibiotic selection was started 36 h post-transduction by adding 2 μg/mL puromycin (Sigma, P8833) and cells were expanded under puromycin selection for 5 days.

## TruHD cell culture

TruHD-Q43Q17M cells were cultured in MEM supplemented with 15% FBS and 1% Glutamax and grown under 4% $O_2$ and 5% $CO_2$ at 37 °C in a 10 cm plate[27]. At 95% confluence, cells were transfected through Lonza nucleofection using the SG Cell Line 4D-Nucleofector Kit. Growth media was replaced 24 h postnucleofection. 5 days post-nucleofection genomic DNA was extracted with PureLink Genomic DNA Mini Kit (Invitrogen). The *HTT* locus was amplified via touchdown PCR using optimized primers (Supplementary Table 3), and submitted for Amplicon-EZ NGS sequencing (Genewiz).

## Homology modeling

Structural models of SpRYc were generated using the SWISS-MODEL server[29], using the PDB 4UN3 (https://www.rcsb.org/structure/4un3) DNA substrate bound Cas9 model as template[6]. Modeled sidechains and loops were curated and adjusted manually using COOT software[45].

## Statistical analysis

Data are shown as the mean of all sample replicates. Data was plotted using Matplotlib and the Prism GraphPad software. For samples performed in independent nucleofection triplicates ($n = 3$) ± SD, statistical significance was determined by two-tailed Student's $t$ test. Calculated $p$ values are represented as follows: *, $p < 0.05$; **, $p < 0.01$; ***, $p < 0.001$; ****, $p < 0.0001$; ns, not significant.

## Reporting summary

Further information on research design is available in the Nature Portfolio Reporting Summary linked to this article.

## Data availability

All data needed to evaluate the conclusions in the paper are present in the paper and supplementary tables. Sequence data that support the findings of this study are available via the NIH Sequence Read Archive via BioProject: PRJNA1019291. Raw data underlying graphical figures are provided as a Source Data file. All additional data can be found at the following Zenodo repository: 10.5281/zenodo.8305744. The SpRYc-ABE8e (#208336) and SpRYc-BE4Max (#208340) plasmids have been deposited to Addgene. Source data are provided with this paper.

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

## Acknowledgements

The research was supported by institutional startup funds to the lab of P.C. from Duke University. GUIDE-Seq work done by N.A. and T.C.R. was supported by a grant (GM115911) to E.J.S from the U.S. National Institutes of Health (NIH). Homology modeling work done by M.P. and M.J. was supported by Swiss National Science Foundation Grant 31003A-182567 (to M.J.). R.T. and N.S. are supported by a grant from the Krembil Foundation. M.J. is an International Research Scholar of the Howard Hughes Medical Institute (HHMI) and Vallee Scholar of the Bert L. & N. Kuggie Vallee Foundation. R.A.S. and B.P.K. acknowledge support from a Natural Sciences and Engineering Research Council of Canada Post-doctoral Fellowship (567791), an MGH Executive Committee on Research Howard M. Goodman Fellowship, and National Institutes of Health (NIH) grants P01-HL142494 and DP2-CA281401. Additional research support was provided by the CHDI Foundation and the Rett Syndrome Research Trust.

## Author contributions

L.Z., S.R.T.K., L.H., V.Y., E.T., and T.S. built constructs, conducted transfections, carried out genome editing experiments, and performed data analyses. N.A. and T.C.R. carried out GUIDE-Seq experiments and analysis. R.S. conducted HT-PAMDA experiments. C.P. and N.S. conducted genome editing assays in TruHD cells. M.P. and M.J. performed structural modeling of SpRYc. M.R.P. conducted in silico docking experiments. L.Z. and C.K. developed cell lines. P.C., L.Z., and S.R.T.K. wrote the paper. G.M.C. supervised cell line construction work. R.T. supervised Huntington Disease work. M.J. supervised homology modeling and structural analyses. B.P.K. supervised HT-PAMDA experiments. E.J.S. supervised GUIDE-Seq experiments. N.J. and J.M.J. provided critical insight and ideas. All authors reviewed and edited the paper. P.C. conceived, designed, directed, and supervised the study.

## Competing interests
