## [Peer Review File · Nature Communications]

Reviewers' Comments:

Reviewer #1:

Remarks to the Author:

The PAM sequence determines the accessibility of Cas proteins of CRISPR/Cas system to DNA sequences. In this study, Koseki et al. utilized protein engineering to fuse the PAM-interacting domain (PID) of SPRY protein to the C-terminus of Sc⁺⁺ protein (which recognizes NNG PAM) to create a new chimeric protein, SPRYc. Firstly, they verified the PAM binding characteristics of SPRYc through the PAM-SCANR bacterial screening system and found that SPRYc has a more efficient binding ability to adenine at position 2 without any bias towards any specific base. Then, using HT-PAMDA bacterial screening system, they found that SPRYc has a broader PAM recognition and DNA cleavage ability compared to other Cas9 proteins except SPRY. Subsequently, they demonstrated that SPRYc could perform targeted cleaving and base editing at target genomic sites with minimal dependence on specific PAMs in HEK293T cells. Moreover, using GUIDE-Seq, they explored that SPRYc had less off-target and improved mismatch tolerance compared to SPRY. Finally, the SPRYc-mediated base editing tool (including ABE8e and BE4max) showed potential clinical value in Rett Syndrome (RTT) and Huntington's Disease (HD) cell disease models. Taken together, the chimeric SPRYc protein developed in this study can recognize NNN PAM and has specific and efficient editing efficiency, with potential therapeutic applications, providing useful tools for genome editing. Nevertheless, I would like to see the following points to be addressed.

Concerns/Suggestions:

1. Using PAM-SCANR bacterial screening, the authors demonstrated SPRYc exhibits a stronger binding affinity towards the second position A of the PAM. However, the cleaving and base editing outcomes observed in HEK293T cells suggest that SPRYc performs better with the C base, which requires further discussion.
2. More gene loci should be tested to characterize the PAM of SPRYc in Figure 2A and B.

Reviewer #2:

Remarks to the Author:

In this work, Koseki and colleagues splice the SpRY and Sc⁺⁺ Cas9 nucleases to generate a hybrid (SpRYc) that exhibits similar broad PAM recognition as SpRY but with a lower propensity for off-targeting. The authors provide further data suggesting deviations in PAM recognition between SpRY and SpRYc and apply SpRYc to generate therapeutically relevant edits for Rett Syndrome and Huntington's Disease. Finally, they provide structural modeling suggesting a mechanism by which SpRYc can accept more flexible PAMs. Overall, the work provides a new step in the engineering of Cas9 nucleases, with SpRYc being relevant for both basic research and clinical development.

Major comments:

1. The substantive indel formation by SpCas9 with the NCG or NCT PAMs is disconcerting since neither (especially NCT) should be recognized as PAMs. This could indicate that the setup generates indels too readily (e.g., due to the extended transient transfection time), resulting in an overly broad picture of indel formation. The base editing data better follows expectations for SpCas9 at least.
2. The authors provided two examples in which SpRYc could be applied to generate therapeutically relevant edits, and in one instance (MECP2) Sp could not generate the desired edit. While this is a reasonable demonstration that the edits can be made by SpRYc, it would be more compelling if SpRY was interrogated with MECP2 and both Sp and SpRY were interrogated with HTT.

Other comments:

3. Can the authors confirm that the SI tables contain all guide sequences used throughout the work? If all are accounted for, I would recommend providing figure/panel citations to make the links clear.

4. P. 2: "...with highly orthogonal PAM targeting." Is this comparing SpRYc to Sp and SpRY? If so, the differences between SpRY and SpRYc were not as strong or as deeply explored. I would recommend softening this claim.
5. P. 2: "Our results would suggest that while...." Some slight rephrasing is in order to make it clear that the underlying individual sequences weren't interrogated, so there might be context for the preference for A at position 2—at least based on the work presented up to that point.
6. P. 3: "Our results reveal that SpRYc-ABE8e can base edit...." To help the reader, I would recommend calling out specific datasets in Fig. 2B supporting this broader conclusion.
7. P. 3: "...with slightly lowered on-target activity (Figure 2D)." By eye, I'm not seeing a difference in on-target activity. Could this be replotted as bar graphs in the SI?
8. P. 3: I recommend adding a citation for HT-PAMDA.
9. Figure 2A: I recommend replacing "DNA modifications" with "indel formation", as DNA modifications strikes me as too general and could encapsulate other types of edits.
10. Figure 3ii: I recommend also illustrating the target and PAM within this sequence so it's clear where the sgRNA is targeting.
11. I couldn't find a citation for Figure S4 nor figure out its relevance to the work. Is it evaluating the potential for self-targeting? If so, this would be a relevant question for a such a PAM-relaxed mutant.

NCOMMS-23-08438 - Response To Reviewers

Reviewer 1

“Using PAM-SCANR bacterial screening, the authors demonstrated SPRYc exhibits a stronger binding affinity towards the second position A of the PAM. However, the cleaving and base editing outcomes observed in HEK293T cells suggest that SPRYc performs better with the C base, which requires further discussion.”

We thank the reviewer for this observation. We have conducted testing of an extensive number of endogenous gene loci with SpRYc, and demonstrate that indeed SpRYc has a slight favoritism to A in the second position, but can generate robust edits at all of the different bases at position 2. Additional data can be found in Supplementary Figure 2.

“More gene loci should be tested to characterize the PAM of SPRYc in Figure 2A and B.”

We have conducted additional extensive genomic loci testing with the SpRYc-ABE8e construct and the results can be found in Supplementary Figure 2.

Reviewer 2

“The substantive indel formation by SpCas9 with the NCG or NCT PAMs is disconcerting since neither (especially NCT) should be recognized as PAMs. This could indicate that the setup generates indels too readily (e.g., due to the extended transient transfection time), resulting in an overly broad picture of indel formation. The base editing data better follows expectations for SpCas9 at least.”

We thank the reviewer for the comment. We agree that the base editing data follows the expectations better, and add additional base editing SpRYc-ABE8e data in Supplementary Figure 2.

“The authors provided two examples in which SpRYc could be applied to generate therapeutically relevant edits, and in one instance (MECP2) Sp could not generate the desired edit. While this is a reasonable demonstration that the edits can be made by SpRYc, it would be more compelling if SpRY was interrogated with MECP2 and both Sp and SpRY were interrogated with HTT.”

We thank the reviewer for suggesting these important controls. We have now redone the entire MECP2 and HTT experiments in relevant cell lines with Sp, SpRY, and SpRYc, and demonstrate that SpRYc can generate therapeutically-relevant edits as well as SpRY on these challenging targets.

“Can the authors confirm that the SI tables contain all guide sequences used throughout the work? If all are accounted for, I would recommend providing figure/panel citations to make the links clear.”

We thank the reviewer for this recommendation. All figure panels and text include a reference to Supplementary Table 1, where we have the complete list of guide sequences.

P. 2: "...with highly orthogonal PAM targeting." Is this comparing SpRYc to Sp and SpRY? If so, the differences between SpRY and SpRYc were not as strong or as deeply explored. I would recommend softening this claim."

We thank the reviewer for this concern. We removed "highly" and softened the statement as following "we combine Sc++ and SpRY to engineer a chimeric Cas9 enzyme that can induce edits with orthogonal PAM targeting".

P. 2: "Our results would suggest that while...." Some slight rephrasing is in order to make it clear that the underlying individual sequences weren't interrogated, so there might be context for the preference for A at position 2—at least based on the work presented up to that point.

We thank the reviewer for this suggestion. We have now indicated that the results are based on "aggregate Sanger sequencing analysis."

"Our results reveal that SpRYc-ABE8e can base edit...." To help the reader, I would recommend calling out specific datasets in Fig. 2B supporting this broader conclusion.

We thank the reviewer for this recommendation. We have added an additional sentence to help call the reader's attention to a few of the specific PAMs where SpRYc-ABE8e most outperforms SpRY-ABE8e.

P. 3: "...with slightly lowered on-target activity (Figure 2D)." By eye, I'm not seeing a difference in on-target activity. Could this be replotted as bar graphs in the SI?

We thank the reviewer for this suggestion. We have replotted the heat map data as a bar graph and included it in SI as Supplementary Figure 5 for additional clarity.

P. 3: I recommend adding a citation for HT-PAMDA.

Thank you for this recommendation. We added the citation for HT-PAMDA as reference #23.

Figure 2A: I recommend replacing "DNA modifications" with "indel formation", as DNA modifications strikes me as too general and could encapsulate other types of edits.

Thank you for this suggestion. We replaced the "DNA modifications" label with "indel formation" throughout the whole manuscript.

Figure 3ii: I recommend also illustrating the target and PAM within this sequence so it's clear where the sgRNA is targeting.

Thank you for this suggestion. We have re-made Figure 3C to be a bar plot comparing Sp, SpRY, and SpRYc on editing the *HTT* repeat, and we illustrate the target and PAM within the context of the *HTT* repeat region.

I couldn't find a citation for Figure S4 nor figure out its relevance to the work. Is it evaluating the potential for self-targeting? If so, this would be a relevant question for a such a PAM-relaxed mutant.

We thank the reviewer for recognizing this oversight. Yes, Figure S4, now Figure S5, is meant to demonstrate no self-targeting of the gRNA cassette, which is a potential complication of a PAM-relaxed mutant. We now cite this figure in the manuscript under the mechanism results section.

Reviewers' Comments:

Reviewer #1:

Remarks to the Author:

In this revised manuscript, the authors tested A-to-G editing at more endogenous loci and confirmed that SpRYc performs better with the A base at position 2 of PAM. They also generated considerable edits at all of the different bases. Which well demonstrated the superiority of base C over A that we observed in the first manuscript. Taken together, the authors well addressed most of points raised by the reviewers. The manuscript has been substantially improved.

Reviewer #2:

Remarks to the Author:

The authors have taken reasonable steps to address all reviewers' comments. My only quick suggestion is to add a reference to Table S1 for guide sequences and targets so the interested reader can figure out which sites gave rise to the plotted editing frequencies.

NCOMMS-23-08438A - Response to Reviewers

Reviewer 1

“In this revised manuscript, the authors tested A-to-G editing at more endogenous loci and confirmed that SpRYc performs better with the A base at position 2 of PAM. They also generated considerable edits at all of the different bases. Which well demonstrated the superiority of base C over A that we observed in the first manuscript. Taken together, the authors well addressed most of points raised by the reviewers. The manuscript has been substantially improved.”

We thank the reviewer for all of their helpful comments throughout the course of this revision!

Reviewer 2

“The authors have taken reasonable steps to address all reviewers’ comments. My only quick suggestion is to add a reference to Table S1 for guide sequences and targets so the interested reader can figure out which sites gave rise to the plotted editing frequencies.”

We thank the reviewer for their positive, helpful, and insightful comments throughout the revision! We have now included multiple references to the Supplementary Table 1 and the other sequence tables so readers can map editing frequencies back to sequences.